# NEURAL PROGRAM SYNTHESIS BY SELF-LEARNING

## ABSTRACT

Neural inductive program synthesis is a task generating instructions that can produce desired outputs from given inputs. In this paper, we focus on the generation of a chunk of assembly code that can be executed to match a state change inside the CPU and RAM. We develop a neural program synthesis algorithm, *AutoAssemblet*, learned via self-learning reinforcement learning that explores the large code space efficiently. Policy networks and value networks are learned to reduce the breadth and depth of the Monte Carlo Tree Search, resulting in better synthesis performance. We also propose an effective multi-entropy policy sampling technique to alleviate online update correlations. We apply AutoAssemblet to basic programming tasks and show significant higher success rates compared to several competing baselines.

## 1  INTRODUCTION

Program synthesis is an emerging task with various potential applications such as data wrangling, code refactoring, and code optimization (Gulwani et al., 2017). Much progress has been made in the field with the development of methods along the vein of *neural program synthesis* (Parisotto et al., 2016; Balog et al., 2017; Bunel et al., 2018; Hayati et al., 2018; Desai et al., 2016; Yin & Neubig, 2017; Kant, 2018). Neural program synthesis models build on the top of neural network architectures to synthesize human-readable programs that match desired executions. Notice that neural program synthesis is different from *neural program induction* approaches in which neural architectures are learned to replicate the behavior of the desired program (Graves et al., 2014; Joulin & Mikolov, 2015; Kurach et al., 2015; Graves et al., 2016; Reed & De Freitas, 2015; Kaiser & Sutskever, 2015).

The program synthesis task consists of two main challenges: 1). *intractability of the program space* and 2). *diversity of the user intent* (Gulwani et al., 2017). Additional difficulties arise from general-purpose program synthesis including *data scalability* and *language suitability*. We will briefly describe these challenges below.

**Program Space.** The program synthesis process consists of sequences of code with combinatorial possibilities. The number of possible programs can grow exponentially with the increase of the depth of the search and the breath of hypothesis space.

**User Intent.** To interpret and encode user's intention precisely is essential to the success of program synthesis. While writing formal logical definition is usually as hard as defining the program itself, natural language descriptions nevertheless introduce human bias and ambiguity into the process. Our work here focuses on *Inductive Program Synthesis* approach where input-output pairs are demonstrated as a program task specification.

**Scale of Data.** The success of recent neural-network-based methods is often built on the top of large-scale-labeled data. Collecting non-trivial human-written programs/code with task specifications can be expensive, especially a large degree of diversity in the code is demanded. An alternative is to exhaust the entire program space, but it is difficult to scale up without taking exploration priority. We instead show an efficient data collection procedure to explore the program space.

**Choice of Language.** The choice of the programming language is also very important. High-level languages like Python and C are powerful but they can be very abstract. Methods like abstract syntax tree (AST) (Hayati et al., 2018) exist but a general solution is still absent. The query language has also been chosen as a topic of study (Balog et al., 2017; Desai et al., 2016), but it has a limited

domain of application. Here, we focus on a subset of x86 assembly language, which is executed on CPU with RAM access.

**Our Work.** We develop a neural program synthesis algorithm, *AutoAssemblet*, to explore the large-scale code space efficiently via self-learning under the reinforcement learning (RL) framework. During the RL data collection period, we re-weight the sampling probability of tasks based on the model's past performance. The re-weighting sampling distribution encourages the networks to explore challenging and novel tasks. During the network training period, the deep neural networks learn policy function to reduce the breath of the search and the value function to reduce the depth of the search. The networks are integrated with Monte Carlo Tree Search (MCTS), resulting in better performance (Coulom, 2006; Browne et al., 2012). This allows our model to solve more complex problems with over $22^3$ hypothesis space done in 10 steps, while the previous related works considered searching 34 output choices finished in 5 steps (Balog et al., 2017).

The reward for the program synthesis task in RL is very sparse, where usually only very few trajectories reach the target output. Empirically, we find that RL policy network training unsatisfactory by receiving low rewards. This is the result of strong correlations of online RL updates, which lead to non-stationary distribution of observed data (Mnih et al., 2016; Volodymyr et al., 2013). In this paper, we propose a conceptually simple multi-entropy policy sampling technique that reduces the updating correlations.

Since our algorithm can learn from its trails without human knowledge, it avoids the data scalability problem. We test its performance on human-designed programming challenges and observe higher success rates comparing to several baselines.

This paper makes the following contributions:

1. Adapt self-learning reinforcement learning into the neural inductive program synthesis process to perform assembly program synthesis.
2. Devise a simple multi-entropy policy sampling strategy to reduce the update correlations.
3. Develop AutoAssemblet for generating assembly code for 32-bit x86 processors and RAM.

## 2 RELATED WORK

We first provide a brief overview of the Inductive Program Synthesis (IPS) literature. Then, we discuss related works along several directions.

**Inductive Program Synthesis (IPS)** views the program synthesis task as a searching process to produce a program matching the behavior with the demonstrated input-output example pairs (Balog et al., 2017). Given enough training examples, a typical IPS model first searches for potential matching programs, followed by picking the best solution using a ranking mechanism. (Balog et al., 2017) focuses mainly on the search part of the algorithm.

**Neural IPS** Recent progress has been made for neural inductive program synthesis (Menon et al., 2013; Parisotto et al., 2016; Balog et al., 2017; Kalyan et al., 2018). When performing the task in this way, a network learns a probability distribution over the code space to guide the search for matching the given input-output pairs. Despite the observation of promising results, challenges remain in neural program synthesis for e.g. effectively exploring the code space. Usually, such a space exploration is carried out by collecting human-labeled data or performing an enumerate search over program space. Our main difference to the existing method is in the data generation procedures. Menon et al. (2013) learns from a small dataset collected by humans, which is hard to scale in practice. While query language in Balog et al. (2017) can potentially yield millions of programs to train deep networks, the data generation process is done by enumerating search strategies without learning a priority. Our model instead efficiently explores a considerably large space by using a self-learning reinforcement learning strategy.

**Representations of Program State** Balog et al. (2017); Kalyan et al. (2018) focuses on query languages with high-level functions like FILTER and MAP. Gulwani (2011) is a practically working application used in a string programming setting. Hayati et al. (2018) synthesizes high-level source code from natural language. Recently, Graph Neural Networks are seen as a potential solution to encoding complex data structure for high-level languages (Allamanis et al., 2018). In this paper,

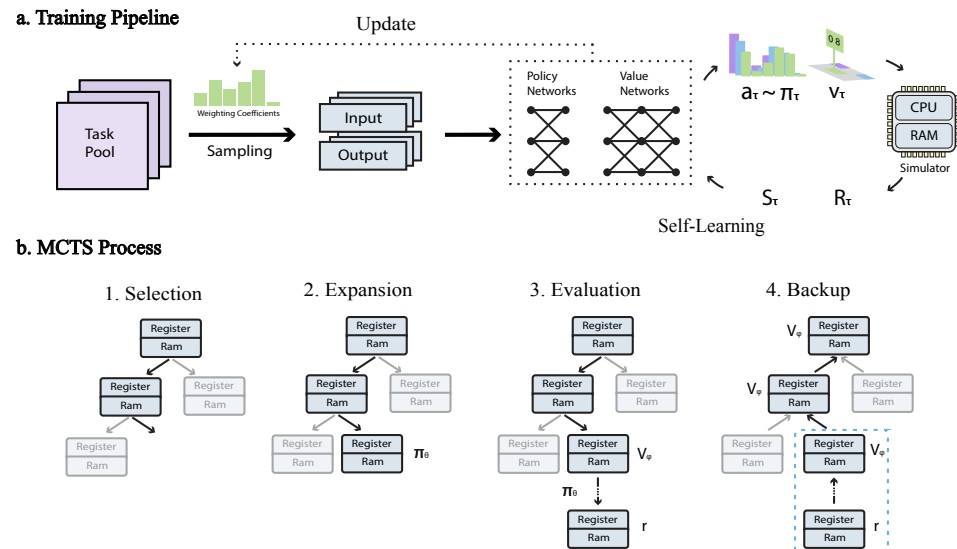

Figure 1: Self-Learning Reinforcement Learning in AutoAssemblet: **a. Training Pipeline** is composed of self-Learning and with re-adjustment of sampling strategy **b. Monte Carlo Tree Search(MCTS)** searches for best policy by utilizing policy networks and value networks.

we provide a different paradigm for Neural IPS by using a subset of x86 instruction set, which generates a program in low-level assembly language. x86 instruction set is widely used in modern computers (Shanley, 2010). Ensemble language sythesis is also studied in Chen et al. (2018). A direct advantage is that the algorithm can synthesize program and receive the reward in the x86 simulator environment. Another advantage is x86 operations directly operate in CPU and RAM in a prescribed format, which is suitable for neural networks to observe the correlation.

# 3    AUTOASSEMBLET

In this section, we first reformulate program synthesis task as an RL problem and introduce necessary notations. Then, we describe the task sampling and multi-entropy policy sampling strategies in the data collection process. Following that, we explain the training procedure of policy and value networks. Finally, we illustrate how Monte Carlo Tree Search utilizes policy networks and value networks during search stage. The full procedure is illustrated in Algorithm 1.

## 3.1    TASK SAMPLING

We train our model following a pipeline consisting of several stages. Initially, we create millions of pilot programs $\lambda_p$ by executing some initial policies $\pi_p$. We then construct input-output $IO$ pairs by running $\lambda_p$ on x86 simulator as our initial task pool. In our setting, input $I$ and output $O$ represent the actual value stored in the CPU registers and the RAM.

During the training process, we assume $N$ programming tasks are sampled from the task pool; each consists of a set of $K$ input-output pairs. Our goal is to learn a neural network $\theta$ that can produce a program $\lambda_\theta$ with a consistent behavior given one set of input-output pairs:

$$\lambda_\theta(I_i^k) = O_i^k \quad \forall i \in 1..N, \forall k \in 1..K. \tag{1}$$

Empirically, we find that sampling from the task pool uniformly leads RL policy networks to overfit easy tasks by giving up difficult tasks with no positive reward, resulting in performance decreasing. To motivate networks to learn a more diverse program distribution, we record the success rate for each task sampled from the pool, which is used to maintain a multinomial distribution. The network gets to sample from the tasks it fails to complete more often, and the easy tasks will not be prioritized.

### 3.2 MULTI-ENTROPY POLICY SAMPLING

The first stage of RL training is the data collection process. For each sampling task, the policy networks $\theta$ recursively synthesize next line of code $a_t$ based on a state representation $s_{(I_t,O)}$, $s_t$ in short, at each time step $t$. During the process, $I_{t+1}$ is generated by x86 simulator by executing $a_t$ based on previous state $I_t$. Program terminates at time step $T$ when either output $O$ is reached ($I_T = O$) or maximum steps have been proceeded. The terminal step will receive a reward $r(s_T)$ such that $r(s_T) = 1$ if the output $O$ is reached, and $r(s_T) = 0$ otherwise. All previous states will receive a decayed reward $r_t = \gamma^{T-t} r(s_T)$ accordingly. We collect $(s_t, a_t, r_t)$ pairs for all $N$ sampled task.

During training, we observe that on-policy updates have a high correlation, which converge to a non-stationary process. Various off-policy methods are proposed to alleviate this problem, including storing previous data in a replay buffer and asynchronously executing multiple agents (Riedmiller, 2005; Schulman et al., 2015; Mnih et al., 2016). However, program synthesis can be regarded as a meta-task where a new task is proposed to be solved at each time. Storing previous experience done in other tasks or updating asynchronously doesn't show clear advantage. We take an alternative approach where temperature $\tau$ in the softmax distribution is alternated. It allows us to synchronously execute multiple agents with different entropy distribution in parallel. In the softmax function, temperature $\tau$ is set to 1 by default. A higher $\tau$ leads to a softer probability distribution, and a lower $\tau$ shifts the distribution towards a one-hot-encoded like distribution (Hinton et al., 2015). The change of temperature reduces the correlation among samples by flexibly altering sampling distribution, which is conceptually similar to off-policy sampling techniques discussed above.

### 3.3 POLICY NETWORKS TRAINING

**Imitation Learning**
We first train our model to imitate programs $\lambda_p$ generated by pilot policy $\pi_p$, which is used to construct input-output $IO$ task pairs. Different from imitation learning that collects human behavior, $\lambda_p$ is not necessary as good as expert demonstrations. For examples, $sub\ 3$ following $add\ 4$ can be optimized to $add\ 1$. Therefore, we expect the quality of $\lambda_p$ decreases for longer programs. However, pilot policy $\pi_p$ is useful to guide policy training and provide $\pi_\theta$ with a stable initialization.

The imitation learning objective is simply obtained from cross-entropy. The networks is trained to make predictions that can maximize the likelihood of actions $a$ from $\pi_p(s)$ conditioned on given inputs:

$$\mathcal{L}_{im}(\theta) = -\mathbb{E}_{s_{1:T} \sim \pi_p} \sum_{t=1}^{T} \log \pi_\theta(a = \pi_p(s_t)|s_t) \tag{2}$$

**Policy Gradient**
While imitation learning duplicates predictions as faithful to the target output from $\lambda_p$, it falls into the problem of *program aliasing*: maximizing the likelihood of a single program would penalize many equivalent correct programs, which hurts long-term program synthesis performance (Bunel et al., 2018). Thus, we perform reinforcement learning on top of a supervised model with the policy gradient technique to optimize outcome directly:

$$\mathcal{L}_{rl}(\theta) = -\mathbb{E}_{s_{1:T} \sim \pi_\theta} \sum_{t=1}^{T} \gamma^{T-t} r(s_T) \sum_{t=1}^{T} \log \pi_\theta(a_t|s_t) \tag{3}$$

To take advantage of both side, we combine $\mathcal{L}_{im}$ and $\mathcal{L}_{rl}$ as a hybrid objective, where $\lambda$ decays with the training process:

$$\mathcal{L}_{hybrid}(\theta) = \mathcal{L}_{rl}(\theta) + \lambda \mathcal{L}_{im}(\theta) \tag{4}$$

### 3.4 VALUE NETWORKS TRAINING

The final stage of the training pipeline is the state-value prediction. When performing long-term program synthesis with Monte Carlo Tree Search (MCTS), we usually cannot reach the terminal state when looking head due to poor policy performance and the depth limitation. To estimate the expected state value under given policy $\pi_\theta$, we train a value function $\phi$ to directly predict discounted outcomes would be received for given state:

$$\mathcal{L}(\phi) = -\mathbb{E}_{s_{1:T} \sim \pi_\theta} \sum_{t=1}^{T} (\gamma^{T-t} r(s_T) - V_\phi(s_t))^2 \tag{5}$$

### 3.5 SEARCHING WITH POLICY AND VALUE NETWORKS

Once trained, we combine policy networks and value networks with a Monte Carlo Tree Search (MCTS) to provide a look-ahead search. Policy network is used to narrow down the breadth of the search to high-probability actions, while the value network is used to reduce the depth of the search with state value estimation.

The MCTS method is composed of 4 general steps: 1) *Selection.* MCTS starts from root state $S_r$ (also as current state $S_c$) and searches for the first available non-expanded leaf state $S_l$ unless the terminal state $S_t$ is found. If the leaf node is found, MCTS will proceed expansion step. Otherwise, it picks new current node $S_c$ from lists of $S_l$ based on four factors: state reward $R_c$, its visit time $N_c$, its parent visit time $P_c$ and exploration factor $\epsilon$.

$$S_c^* = \underset{S_c \in [S_l]}{\arg\max} R_c/N_c + \epsilon \sqrt{2log(P_c)/N_c} \tag{6}$$

2) *Expansion.* After MCTS decides which state $S_c$ to be expanded, it applies policy network $\pi_\theta(a_c|s_c)$ to sample one action from hypothesis distribution, and proceeds to $S_{c+1}$. A good policy network can significantly boost search tree efficiency by reducing the necessary breadth of the expansion. 3) *Evaluation.* We use the same policy $\pi_\theta$ as our rollout policy to search terminal state $S_t$ from node $S_{c+1}$ under $t$ maximum step. If target state is not reached due to code error or maximum step is reached, we use a value function $V_\phi(s_t)$ to estimate terminal state's future outcome. 4) *Backup.* The result of the rollout process is used to update reward for the nodes on the path from $S_c$ to $S_r$.

---

**Algorithm 1** AutoAssemblet Training Process

---

1: **Require**: policy network $\pi_\theta$; value network $V_\phi$; a task pool $S = s_1, ..., s_K$ where $s_k$ is constructed by input state $I_k$ and output state $O$ at time step t; a list of policy temperature $\tau = \tau_1, ..., \tau_M$
2: Initialize the task weighting coefficients $\{w_k\}$ by setting $w_k = 0$ for $k = 1, ..., K$
3: **while** AutoAssemblet has not converged **do**
4:     $p_k = Softmax(w_k)$ // Multi-Entropy Policy Sampling
5:     Sample a batch of tasks $S_B = s_1, ..., s_N$ from task pool $S$ based on $\{p_k\}$
6:     **for** $i$ in $M$ **do**
7:         Collect $(s_t, a_t, r_t)$ pairs based on policy network $\pi_\theta^i$
8:         $w_k \leftarrow w_k + (-1)^{\mathbb{1}\{\pi_\theta^i(I_k) \neq O\}}$ // Update weighting coefficients
9:     **end for**
10:    Update $\pi_\theta$ by Eq. (4) // Policy Networks Training
11:    Update $V_\phi$ by Eq. (5) // Value Networks Training
12: **end while**

---

## 4 EXPERIMENTS

In this section, we describe the results from two categories of experiments. In the first set of experiments, we demonstrate how AutoAssemblet may improve upon the performance achieved through only imitation learning or REINFORCE. In the second set of experiments, we demonstrate the performance of AutoAssemblet on human-designed program benchmarks comparing to other baselines.

### 4.1 EXPERIMENT SETUP

**Action Space** The action space the network searches on is a reduced syntax of the x86 assembly language. We use 4 CPU registers ($\%eax$, $\%ebx$, $\%ecx$, $\%edx$ ), and 4 main data transfer / integer instructions ($addl$, $subl$, $movl$, $imull$), 10 digit numbers, and optionally 4 RAM positions ($-0(\%rbp)$, $-4(\%rbp)$, $-8(\%rbp)$, $-12(\%rbp)$ ).

**Observation Space** The observation space the networks look at is the current values in CPU register and RAM as well as target values. The network takes in the observation as input and generates the next line of code. During the training and searching process, the instruction is then compiled and executed by CPU (or a simulator of CPU) to get subsequent state.

**Model** Our policy networks mainly consist of two parts: a task encoder and a program decoder. The task encoder consists of an embedding layer and five fully-connected (FC) layers followed by $tanh$. The start state and target state pairs are concatenated and feed into the embedding layer. We implemented an unrolled RNN as the code generator. The generator receives a context vector from the previous FC network, which extracts information about a task. Since the code length is fixed, the unrolled RNN can significantly increase the training speed. Our value networks mainly consist of two parts: a task encoder and a value prediction layer. The task encoder shares the same architecture with the policy networks, and the value prediction layer is an additional FC layer followed by a sigmoid function. Potentially, the task encoder between policy networks and value networks can be shared.

### 4.2 PERFORMANCE COMPARISON

We trained neural networks on different augmentations of the training process to demonstrate the success of our method in different settings. The hyper-parameters we tested were the size of the task pool, the lines of code in the training code, the size of the input-output set, and the number of registers. For all our experiments in this category, we used the default hyper-parameters of using a pair of instances in the input-output set, 3 lines of code, 4 registers, and 300,000 coding tasks. We compare our results with ones learned from imitation learning and REINFORCE. The prediction accuracy is evaluated on a hold-out validation set.

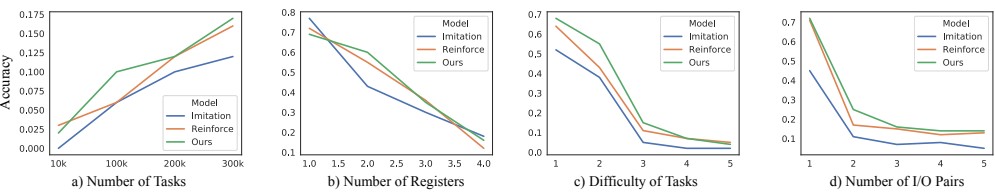

Figure 2: Performance comparison under four experiments setting

**Increase of search difficulty.** In our first experiment shown in Figure 2a), we evaluated our models on increasing task pool sizes and tested for the accuracy of the generated program. Notice, both results from imitation learning and REINFORCE drop after task pool scales larger. This is largely due to the policy learned by networks overfits to simple and redundant tasks. Our *AutoAssemblet* model shows more robust improvements with high data efficiency.

Additionally, in Figure 2b), the search space is largely increased by the number of registers used, which leads to the sharp decrease in accuracy, but our model has the most gentle slope.

**Increase of task difficulty.** As our third experiment shown in Figure 2c), we increase the task difficulty by increasing the number of steps pilot program used to generate input-outputs. For single step task, our model shows worse performance comparing to another two baselines, but it indicates significant higher accuracy when task challenge increases. We hypothesize that because the MCTS is unnecessary searching for a multi-line solution, thus not recognizing the simple single-line solution. For tasks that generate from 5 steps, it boosts accuracy from 2% to 12% compare to imitation learning. This is because our model does not suffer from overfitting the memorization of the exact data pattern.

Figure 2d) demonstrates that neural network's performance decreases when adding more input-output pairs to the training data. The results are more consistent after increasing to more than two input-output pairs. We hypothesize the reason for this is that having a single input-output pair makes the task of finding the mapping between the input and output ambiguous. For a single input-output pair, multiple high-level abstraction could solve such mapping. However, increasing the number of pairings to at least 2 seems to remove this ambiguity for more instances within the training data.

### 4.3 HUMAN-DESIGNED PROGRAMMING CHALLENGES

Previously, we trained the neural network on self-explored tasks generated by the pilot policy. In this experiment, we designed a set of human-designed tasks to test its potentials to solve meaningful programming tasks. As a note, we create two input-output pairs per task to ensure that the network produces the desired program.

We divided the tasks into three categories based on their level of abstraction. Within the easy task set, we have the tasks of addition, subtraction, multiplication of registers, and finding the minimum and maximum values. Within the medium task set, we have the tasks of moving the value of a register to another register location, adding and subtracting a value larger than 10, and adding a constant to all registers. Adding or subtracting a value greater than ten may be difficult because this is a state that neural network rarely experienced during training. Lastly, the hard task set includes high-level abstraction, such as filtering, sorting, switching registers, and finding the two maximum or minimum values. Respectively, there are fifty, forty, forty human-designed tasks within the easy, medium, and hard test sets.

| Model | Imitation | REINFORCE | MCTS(PV) | AutoAssemblet (Ours) |
|---|---|---|---|---|
| **EASY** | | | | |
|    SUCCESS RATE | 60% | **64%** | 63% | 59% |
|    AVE STEPS | 2.7 | 3.1 | 2.5 | 2.9 |
| **MEDIUM** | | | | |
|    SUCCESS RATE | 25% | 29% | 35% | **37%** |
|    AVE STEPS | 4.5 | 7.5 | 9.2 | 6.9 |
| **HARD** | | | | |
|    SUCCESS RATE | 2% | 5% | 5% | **10%** |
|    AVE STEPS | 13.1 | 12.4 | 13 | 12.5 |
| **TOTAL** | | | | |
|    SUCCESS RATE | 34.4% | 38.2% | 40.2% | **40.4%** |

Table 1: Results of Human-Designed Programming Challenges. Details for tasks designed in each difficulty level is explained at A.1

The similar performance between the model learned from imitation learning and REINFORCE reflects the limitations of only using a policy network for code generation, and thus suggests that improvement can come from the use of a value function.

Our model performs better than all baseline models with an over 20% overall improvement over the nearest models. As seen in table 1, our model achieved a higher accuracy for test set within different difficulty level.

## 5 DISCUSSION

### 5.1 WHAT AUTOASSEMBLET LEARNED FROM NETWORKS

In this section, we highlight a set of programs generated by AutoAssemblet to solve the human-designed tasks we created. These examples illustrate the model's ability to create low-level code that is interpretable by human and to find shortcuts of the problems.

First, the model can learn a continuous representation of number system, so it can learn simple algebra to switch a sequence of number to another sequence. Second, for those digits not frequently seen in training data (which only appears sparsely), the model learns to approach the value with more efficient operation (such as multiplication), and to finetune the value by add or subtract small

value to get to the exact target. Third, the mapping between register name and value is also learned by the model. It knows which position should be changed and put the corresponding register name to generated instruction to achieve its goal, rather than changing digits randomly.

We also try the setting including RAM. The input is originally stored in RAM, and the target is only about transforming the values in RAM. As a simple problem, the network learns to first load the value to register, apply operations on it, and store it back to memory after the task is done. We present several demo programs below:

**Algebra:**
```
imull %eax, %ecx
addl  $2,    %ecx
```

**Input-output example:**
*Input*:
```
[5, 1, 7, 8]
[4, 3, 7, 0]
```
*Output*:
```
[5, 1, 37, 8]
[4, 3, 30, 0]
```

*Description*:
The task is to add 30 to the third register. It tests the model's ability to perform simple arithmetic. Instead of addition, the model utilizes multiplication.

**MAP:**
```
addl $1, %ebx
addl $1, %edx
addl $1, %eax
addl $2, %ecx
subl $0, %ecx
subl $1, %ecx
```

**Input-output example:**
*Input*:
```
[8, 1, 0, 7]
[2, 4, 5, 7]
```
*Output*:
```
[9, 2, 1, 8]
[3, 5, 6, 8]
```

*Description*:
The goal is to add 1 to each register, which would show up in a human-made program of iteration.

**Filter:**
```
subl $1, %ecx
subl $2, %eax
addl $4, %edx
subl $4, %edx
addl $4, %edx
```

**Input-output example:**
*Input*:
```
[1,5,0,2]
[1,8,0,2]
```
*Output*:
```
[-1,5,-1,-1]
[-1,8,-1,-1]
```

*Description*:
Filters out the values less than 3 and sets them to -1.

**Sort:**
```
addl $4, %ebx
subl $4, %eax
```

**Input-output example:**
*Input*:
```
[5,1,7,8]
```
*Output*:
```
[1,5,7,8]
```

*Description*:
Sort the variables in the list and return. Sorting is highly dependent on the input, therefore we only sort a single list of inputs.

<table>
<tr><td>

**Load/Store Memory:**
```
movl -8(%rbp), %ebx
subl $3,        %ebx
movl %ebx,  -4(%rbp)
```

</td><td>

**Input-output example:**
*Input*:
```
CPU:[0, 0, 0, 0]
RAM:[2, 8, 0, 1]
CPU:[0, 0, 0, 0]
RAM:[6, 5, 4, 9]
```
*Output*:
```
CPU:[0, -3, 0, 0]
RAM:[2, -3, 0, 1]
CPU:[0, 1, 0, 0]
RAM:[6, 1, 4, 9]
```

</td><td>

*Description*:
Load a variable from RAM, add it with a number, and store it to RAM again.

</td></tr>
</table>

<table>
<tr><td>

**Summation:**
```
addl $3, %edx
addl $4, %ebx
subl $4, %ebx
addl %ebx, %edx
addl %eax, %edx
```

</td><td>

**Input-output example:**
*Input*:
```
[0,1,3,2]
[7,4,3,0]
[4,4,3,3]
```
*Output*:
```
[0,1,3,6]
[7,4,3,14]
[4,4,3,14]
```

</td><td>

*Description*:
Find the summation of the registers and set the value as the final register in the input.

</td></tr>
</table>

## 5.2 LIMITATIONS AND FUTURE DIRECTIONS

However, limitations still remain. One major issue is that we prohibit the model to use control-flow instructions. Many challenging tasks can be easily solved with a loop and if statement. Our network decides next instruction only by observing current and target machine states. There is no sufficient mechanism to step back to certain historical states to re-insert a branching clause into instruction flow. Thus an advanced global planning mechanism is necessary to introduce control-flow instructions into our setting. Besides,the abuse of if-else clauses can also hampers extraction of general methods to solve a task, because an agent can be cheated in a simple adversarial example: reduce any multi-instance based problem to one-instance problem by if-else clause and take trivial steps to solve each instance separately. Such limitation contradicts the original purpose of finding the universal method to solve all instances in the same task.

Another challenge is variable management. Variables in high-level language are bounded to certain stack or heap positions in assembly language. The agent needs to know which position is accessible to avoid stack-overflow and segmentation fault, and which variable is available in current scope to avoid getting out-dated value of it. This problem is especially important in solving tasks with variables needs long-term usage. A variable management mechanism should also be introduced to apply accurate operations on variables and to protect the prohibited area in RAM and registers.

## 6 CONCLUSION

We have presented a neural program synthesis algorithm, AutoAssemblet, for generating a segment of assembly code to execute state change in the CPU and RAM. We overcome the limitations in the previous program synthesis literature by designing a self-learning strategy suitable for code generation learned via reinforcement learning. We adapt policy networks and value networks to reduce the breadth and depth of the Monte Carlo Tree Search. Applicability for AutoAssemblet using an effective reinforcement learning approach has been observed in our experiments, where a sequence of assembly codes can be successfully generated to execute the stage changes within the CPU. It points to a promising direction for program synthesis, if properly formulated, to learn to code at scale.

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

## A APPENDIX

### A.1 TASK DESCRIPTION

Description of human-designed programming tasks:

Easy:
1) Summing 2 registers (setting sum value @ first register) (10 tasks)
2) Summing 4 registers (setting sum value @ first register) (10 tasks)
3) mapping: adding 1 to all registers (10 tasks)
4) subtract 15 from second index (10 tasks)
5) add 10 to first index (10 tasks)
6) 51-100: 50 generated examples of 1-line code (10 tasks)

Medium:
1) Adding prior registers (add to register i value from register i-1) (10 tasks)
2) subtracting prior registers (10 tasks)
3) maximum value (to first register) (10 tasks)
4) minimum value (to first register) (10 tasks)
5) switching 1st and 4th register (10 tasks)
6) 51-100: 50 generated examples of 1-line code

Hard:
1) Filtering (keep value if above 5, set others to 0) (10 tasks)
2) sorting (4 registers)(10 tasks)
3) reversing registers (10 tasks)
4) mapping: use register 1 as value for rest of registers (10 tasks)
5) mapping: adding +i for ith register (10 tasks)

## A.2 DATASET DISCRIPTION

We designed 4 experiment settings of different variable:
1. Number of tasks. We use 4 dataset, with 10000, 100000, 200000, 300000 tasks respectively. Default other parameters are 4 registers, 4 operations, 10 numbers, 3 lines of program.
2. Lines of program. We use 4 dataset, with 1, 2, 3, 4, 5 lines of program respectively. Default other parameters are 4 registers, 4 operations, 10 numbers, 300000 tasks.
3. Number of examples per task. We use 4 dataset, with 1, 2, 3, 4, 5 pairs of examples per task respectively. Default other parameters are 4 registers, 4 operations, 10 numbers, 300000 tasks.
4. Number of registers. We use 4 dataset, with 1, 2, 3, 4 registers respectively. Default other parameters are 300000 tasks, 4 operations, 10 numbers.

