# OpenReview forum: "Neural Program Synthesis By Self-Learning"
_ICLR.cc/2020/Conference — Reject_

### Official Review · AnonReviewer3 · 2019-10-23
**Official Blind Review #3**

**Rating:** 3

**Review:**

[Summary]
This paper addresses the problem of synthesizing programs (x86 assembly code) from input/output (I/O) pairs. To this end, the paper proposes a framework (AutoAssemblet) that first learns a policy network and a value network using imitation learning (IL) and reinforcement learning (RL) and then leverages Monte Carlo Tree Search (MCTS) to infer programs. The experiments show that AutoAssemblet can synthesize assembly programs from I/O pairs to some extent. Ablation studies suggest the proposed IL and RL guided search is effective.

Significance: are the results significant? 2/5
Novelty: are the problems or approaches novel? 2/5
Evaluation: are claims well-supported by theoretical analysis or experimental results? 4/5
Clarity: is the paper well-organized and clearly written? 4/5

[Strengths]

*clarity*
The overall writing is clear. The authors utilize figures well to illustrate the ideas. Figure 1 clearly shows the proposed pipeline as well as the MCTS process. In general, the notations and formulations are well-explained.

*technical contribution*
- Optimizing both the imitation learning loss and the reinforcement learning loss yields better performance when more tasks are available and tasks are more difficult.
- Leveraging a learned policy network and value network for improving the efficiency of the MCTS seems effective.

*ablation study*
Ablation studies are comprehensive. The proposed framework first optimizes two losses (IL and RL) and leverages the learned policy network and the value network for improving MCTS. The provided ablation studies help analyze the effectiveness of each of them.

*experimental results*
- All the descriptions of the experiments and the presentations of the results are fairly clear.
- The results demonstrate the effectiveness of the proposed RL guided MCTS.

[Weaknesses]

*novelty*
Overall, I do not find enough novelty from any aspects while the overall effort of this paper is appreciated. The reasons are as follows.
- This  "self-learning" framework is not entirely novel since it has been proposed in [1], where the model is trained on a large number of programs that were randomly generated and tested on a real-world dataset (FlashFill).
- The hybrid objective (IL+RL) has been explored in neural program synthesis [2] (a supervised learning model is fine-tuned using RL), learning robot manipulation [3], character control [4], etc.
- Utilizing Monte Carlo Tree Search for program synthesis has been studied in many works. [5] proposes to treat the network outputs as proposals for a sequential Monte Carlo sampling scheme and [6] presents an RL guided Monte Carlo tree search framework.
- Program synthesis on assembly languages: RISC-V [6], etc.

*related work*
The descriptions of the related work are not comprehensive. While many neural synthesis works [1, 5, 7-10] have explored a wide range of settings for learning to perform program synthesis, they are not mentioned in the paper. I suggest the authors conduct a comprehensive survey on this line of works.

*baselines*
In my opinion, the baselines (imitation, REINFORCE, MCTS) presented in the paper are far from comprehensive. I believe the following baselines should also be considered:
- As the proposed model optimizes a combination of the IL loss and the RL loss, it would make sense to also evaluate a model optimizing this hybrid loss.
- Search-based program synthesis baseline (i.e. learning guided search vs heuristic search)
- Comparing the proposed framework against some neural induction baselines would confirm the importance and effectiveness of explicitly synthesizing programs instead of directly predicting the outcome/output. This has been shown in [1, 7].

*testing set*
The testing sets are extremely small (with only 50, 40, 40 programs), which makes the results less convincing. Also, how those testing sets were created is not mentioned anywhere in the paper. It only states "we designed a set of human-designed tasks".

*number of given I/O pairs*
It is not mentioned anywhere how the authors split the observed I/O pairs and the assessment I/O pairs such as what has been done in most of the works [1, 2, 7, 8]. While the observed I/O pairs are input to the program synthesis framework, the assessment I/O pairs are used to evaluate the synthesized programs. By doing so, the more observed I/O pairs are given, the more accurate the synthesized programs should be (assuming the model can find programs that fit the observed I/O pairs).
- The discovery (Figure 2d) in this paper is contradictory to what is mentioned above: the program synthesis accuracy decreases when more I/O pairs are given. I am assuming the authors do not split observed I/O pairs and assessment I/O pairs.
- With the setup where the observed I/O pairs are separated from the assessment I/O pairs, a larger number of K (the number of the observed I/O pairs) should be used so that it is more likely that the program for each task is unique and the evaluation would make more sense.

[1] "RobustFill: Neural Program Learning under Noisy I/O" in ICML 2017
[2] "Leveraging Grammar and Reinforcement Learning for Neural Program Synthesis" in ICLR 2018
[3] "Reinforcement and Imitation Learning for Diverse Visuomotor Skills" in RSS 2018
[4] "DeepMimic: Example-Guided Deep Reinforcement Learning of Physics-Based Character Skills" in SIGGRAPH 2018
[5] "Learning to Infer Graphics Programs from Hand-Drawn Images" in NeurIPS 2018
[6] "Program Synthesis Through Reinforcement Learning Guided Tree Search" arXiv 2018
[7] "Neural Program Synthesis from Diverse Demonstration Videos" in ICML 2019
[8] "Execution-Guided Neural Program Synthesis" in ICLR 2019
[9] "Learning to Describe Scenes with Programs" in ICLR 2019
[10] "Learning to Infer and Execute 3D Shape Programs" in ICLR 2019

===== After rebuttal =====

I appreciate the authors for the revision and for clarifying some points. I am still not entirely convinced by the response and the revision.

First, I mentioned several papers that are related to this work, the authors failed to discuss the difference between this submission and these works.
- The "self-learning" paradigm is used in [1, 2, 7, 8] but [1, 7] are still not mentioned. If the authors intentionally ignore these works so that they can claim the novelty, it is not acceptable.
- While the proposed hybrid objective is very similar to the one proposed in [2, 3, 4], the revision does not mention this.
- Why [5, 6] that utilizing Monte Carlo Tree Search for program synthesis are still missing from the revision? I believe these works are very relevant to this submission.
- The authors acknowledged that they did not know about [6] that works on program synthesis for an assembly language. Yet, this paper is still missing from the paper.
I spent a lot of time conducting a survey in this field to help the authors to improve this submission and trying to identify the novelty of this submission. However,  the authors just chose to ignore it, which is very disappointing.

Second, many suggestions that I made are just rejected by the authors because they believe it is "not very easy and trivial". Given this self-learning setting, I believe it would be easy to implement some program induction baselines using supervised learning. Or how about search-based program synthesis baselines?

The writing about 50, 40, 40 testing sets is very misleading. I believe all the reviewers just think the testing accuracy was computed using those testing sets.

From my point of view, "program aliasing” does not mean there are not sufficient I/O examples to fully describe a task; Instead, it means there are programs written differently can be semantically the same, which means no matter how many input examples are given, their outputs will always match.

Overall, I believe this submission requires a serious revision and I firmly recommend this paper to be rejected.

**Experience Assessment:**

I have published one or two papers in this area.

**Review Assessment: Checking Correctness Of Derivations And Theory:**

N/A

**Review Assessment: Checking Correctness Of Experiments:**

I carefully checked the experiments.

**Review Assessment: Thoroughness In Paper Reading:**

N/A

---

> ### Author Response · Authors · 2019-11-15
> **Reply to review**
>
>
>        Thanks a lot  for the valuable suggestions and provision of so many reference work.  Our reply to some of the questions above is as follows:
> Novelty:
> Again thanks for providing many literatures. Some of them(like the RISC-V paper) are first seen and really explored our view.
>
> To make Reinforcement Learning methods work well in code generation is not a trivial task. One critical problem in RL training is to deal reward sparseness and update correlation in sampling stage.To point out,
> Our novelty is not introducing reinforcement learning and MCTS, which have been explored in many other similar scenarios. The slight step we move forward is by using task sampling and multiple policy to improve training stableness, forcing the agent to really learn to solve problems rather than circumventing them. We regard this as important because x86 state and action space are non-convex and relatively more complex, so it’s very likely that the area being explored are highly correlated with current policy, which in turn decide the explored area. This meant agent would overfit certain type of tasks instead of finding ad-hoc solutions for different tasks.
>
> The problem, also called distribution shift, can result in worsening performance in RL as time goes by, because RL agent is trained on data sampled by the agent itself. To encourage stability, we encourage RL sample a balanced distribution of tasks by using the technique of policy sampling and task sampling.
>
> To support our assumption, we revised our work by adding another baseline trained without task sampling.The performance will soon drop to less than 1% accuracy compared to 10%+ accuracy on test set without task sampling, because RL agent only choose to sample easy tasks in training time.
>
>
>
> Baseline:
>
> 	Evaluate a model optimizing this hybrid loss: To clarify, the REINFORCE result is done in this setting you suggested, which still uses imitation. That’s because without pretrain on imitation learning, training purely with REINFORCE only will converge with extreme low score due to distribution shift, which we also tried but not presented.
>
> 	Comparing the proposed framework against some neural induction baselines: We are lacking baseline because of our choice for assembly language as our playground, which is a relatively new setting. Reimplement is also not very easy and trivial because not many work published original code, especially for reinforcement learning.
>
> In this situation, we choose to compare between several algorithms implemented ourselves to fit the setting. This is also part of the reason we choose a widely adopted language, behind which might be more training resources from different programs, which can encourage community joint effort in the future.is is also part of the reason we choose a widely adopted language, behind which might be more training resources from different programs, which can encourage community joint effort in the future.
>
>
> Dataset:
> the 50, 40, 40 testset is not used for calculating accuracy. We use a bigger test set of 1000 samples generated randomly same as the process of training. The smaller testset is written manually to test the model’s ability to solve some problems that are easier to interpret. We use it to exhibit the generated program and the corresponding intention within our paper.
>
> Experiment Setting:
>
> Without semantic guidance or training time regularization, we deem every “aliasing program” as correct during test time. As is mentioned in [2](Leveraging Grammar and Reinforcement Learning for Neural Program Synthesis), there would be a problem of “program aliasing” when there is not sufficient I/O examples to fully describe a task, thus many programs would be semantically equivalent, albeit not recover human intention except for only one program. But this literally resulting that less pairs actually leads to better accuracy not because performance improve, but lower threshold. To get a more fair result, we revised our work by testing all experiments with an unseen hold-out I/O pair. Although results from all models drop, AutoAssemblet still outperforms baselines reported.

---

> ### Author Response · Authors · 2019-12-23
> **After rebuttal**
>
> Thanks for providing constructive comments and pointing out the missing references!
>
> During the rebuttal stage, we focused on getting new baseline results and made a quick update near the deadline. However, due to a miscommunication among us, we missed a few references and didn't discuss them all. We are very sorry for the mistake.
>
> We are thoroughly revising the manuscript and will make an update soon. Thank you very much!

---

> > ### Comment · AnonReviewer3 · 2019-12-24
> > **Re: After rebuttal**
> >
> > Thanks for the reply. I look forward to the revised manuscript and do wish the authors all the best for the next submission cycle.

---

### Official Review · AnonReviewer1 · 2019-10-23
**Official Blind Review #1**

**Rating:** 1

**Review:**

The authors propose to tackle the problem of neural inductive program synthesis using a combination of REINFORCE, imitation learning, and MCTS. Furthermore, they propose a method for sampling tasks that aims at minimising task correlation when optimizing the policy parameters. They test their method on a set of input-output sequences provided by automatically generated x86 programs, and a small set of manually designed programs.


Given the current state of the manuscript, this is a clear reject. Some of the main issues I notice are:

1. The novelty of the proposed (combined) method is unclear, given that it is a relatively straightforward combination of relatively simple and battle-tested techniques; I don't consider this in general to be a problem, but previous work has explored the problem way more significantly both algorithmically and in modeling terms.

2. The experimental section is entirely composed of non-standard datasets, and - in general - the manuscript lacks almost entirely in critical details regarding how the datasets are generated; e.g. What is the pilot policy? Is there a finite set of IO tasks defined for all the experiments? How were the manual tasks designed? What are the qualitative differences in task dynamics between the two experiment settings?

3. The baselines are extremely basic, especially considering that there are a multitude of papers - some of which are mentioned in sections 1 and 2 - that would provide for some excellent comparison. Furthermore, there's a lack of details about the model setup, hyperparameters, state featurization, and so on.

4. Across the manuscript there seems to be some apparent confusion on whether they are tackling the problem as a multi-task setting, where each IO set is considered to be a separate task, or whether all of these tasks are just instances of a single MDP. Given that the program space is well defined, I would think that modelling the problem as a single task is more appropriate, however the authors seem to have chosen a multi-task approach. However, in such case there's also a lot of previous work on multi-task RL and meta-learning that should have been mentioned and potentially used / compared against. This also affects how sensible their proposed sampling method is (and whether the assumption wrt on-policyness actually is reasonable).

I would encourage the authors to improve the work in the following ways:
- Please include the nitty gritty details about the setup, including dataset, simulator, training and algorithmic hyperparameters, state featurization, etc. - try to make experimental reproduction as easy as possible exclusively based on the manuscript.
- Clarify the setup wrt. point 4 above; ideally formalize the problem statement using MDPs, such that it can be properly reasoned upon.
- Review more closely previous work, and choose a suitable (and possibly recent and as close to SOTA as reasonably possible) baseline, such that the proposed methods can be quantitatively and qualitatively compared.
- Similarly, please attempt to test your method on existing environments and datasets, such that any analysis against previous baselines can be fairly assessed.


**Experience Assessment:**

I have published in this field for several years.

**Review Assessment: Checking Correctness Of Derivations And Theory:**

I assessed the sensibility of the derivations and theory.

**Review Assessment: Checking Correctness Of Experiments:**

I carefully checked the experiments.

**Review Assessment: Thoroughness In Paper Reading:**

I read the paper at least twice and used my best judgement in assessing the paper.

---

> ### Author Response · Authors · 2019-11-15
> **Reply to review**
>
> Thanks for the valuable suggestions as well as criticism.  Our reply to some of the questions  above is as follows:
>
> 1. Novelty:
> To make Reinforcement Learning methods work well in code generation is not a trivial task. One critical problem in RL training is to deal reward sparseness and update correlation in sampling stage.To point out,
> Our novelty is not introducing reinforcement learning and MCTS, which have been explored in many other similar scenarios. The slight step we move forward is by using task sampling and multiple policy to improve training stableness, forcing the agent to really learn to solve problems rather than circumventing them. We regard this as important because x86 state and action space are non-convex and relatively more complex, so it’s very likely that the area being explored are highly correlated with current policy, which in turn decide the explored area. This meant agent would overfit certain type of tasks instead of finding ad-hoc solutions for different tasks.
>
> The problem, also called distribution shift, can result in worsening performance in RL as time goes by, because RL agent is trained on data sampled by the agent itself. To encourage stability, we encourage RL sample a balanced distribution of tasks by using the technique of policy sampling and task sampling.
>
> To support our assumption, we revised our work by adding another baseline trained without task sampling.The performance will soon drop to less than 1% accuracy compared to 10%+ accuracy on test set without task sampling, because RL agent only choose to sample easy tasks in training time.
>
>
> 2. Dataset:
> Our code dataset is generated by randomly sampling from syntax of assembly language. After generating the generated program, we feed random inputs within our number range into the simulator to get corresponding output of the program. Because the I/O pairs rely on generated program, our dataset differentiate across different experiments in lines of program and choices of registers which is necessary to compose an instruction. During the experiment, we generated 300k tasks, and sample batches tasks from them during training.
>
> During revision, we also trying to make our test dataset more convincing by separate observed and held-out example pairs. We updated new results in our paper.
>
> We consider this is still a single-task setting in machine learning setting, where each programming task can be referred to as a datapoint. Multi-task learning usually refers to the setting introducing more than one learning objective, and it is never the intention of this paper.
>
>
> 3. Baseline:
> We also agree that we are lacking baselines to make our experiment more convincing, but we think that’s because of our choice for assembly language as our playground, which is a relatively new setting. Reimplement is also not very easy because not many work published original code, especially for reinforcement learning. To compensate,  we choose MLE as our baseline,which is also adopted in other paper. (like https://arxiv.org/pdf/1805.04276.pdf). This is also part of the reason we choose a widely adopted language, behind which might be more training resources from different programs, which can encourage community joint effort in the future. We will also explore more literature to find other possibilities.
>
> 4. Details:
> We add some details about our experiment at the end of this file.

---

### Official Review · AnonReviewer2 · 2019-10-24
**Official Blind Review #2**

**Rating:** 3

**Review:**

This paper tackles the problem of program synthesis in a subset of x86 machine code from input-output examples. First, the paper uses a random code generation policy to generate many programs and executes them with random inputs to obtain I/O and program pairs. Then, the paper trains a model using imitation learning on this dataset, and then transitions to to using policy gradient and Monte Carlo Tree Search methods to train the network.

I thought it was quite cool that the paper generates assembly code, but the set of instructions allowed is quite limited. While the paper seems a bit dismissive about prior work such as Balog et al 2017 that use "query languages", the higher-level primitives found in such languages (like "filter") could also mean that the models involved have to learn higher-level semantics than what this model needs.

Furthermore, the paper only uses two input-output examples to specify the desired behavior, and the accuracy of the model's output is only evaluated on that pair of examples. While the paper discusses in Section 5.2 that it is important to learn general methods to solve a provided task, this evaluation setting prevents . Similar to previous work like Bunel et al 2018, I would encourage the authors to measure how well the generated programs can do on held-out example input-output pairs, to see whether the model could successfully recover the "intended" program; to assist with this, we can also increase the number of input-output pairs used to specify the task. Of course, this would make it probably impossible to recover the "hard" problems from section 4.3, since those require conditional execution.

I think the paper should have cited works such as
- https://arxiv.org/abs/1906.04604
- https://openreview.net/forum?id=H1gfOiAqYm
- https://papers.nips.cc/paper/8107-improving-neural-program-synthesis-with-inferred-execution-traces
which also make use of execution information in order to predict the code. In particular, the first paper in this list also uses tree search methods to generate programs as a sequence of loop and branch-free instructions, so I believe it should be quite similar to this paper at a high level.

Considering the above limitations with the evaluation methodology, and the limited novelty of this work in light of these citations, I vote for weak reject.

**Experience Assessment:**

I have published one or two papers in this area.

**Review Assessment: Checking Correctness Of Derivations And Theory:**

I assessed the sensibility of the derivations and theory.

**Review Assessment: Checking Correctness Of Experiments:**

I assessed the sensibility of the experiments.

**Review Assessment: Thoroughness In Paper Reading:**

I read the paper at least twice and used my best judgement in assessing the paper.

---

> ### Author Response · Authors · 2019-11-15
> **Reply to review**
>
> Thank you for your valuable review and suggestions. Our reply to some of the questions above is as follows:
>
> We are also regret for not using big set of instructions from x86, but this problem is not trivial to scale up. For example, we adopted only a limited subset from assembly language because there is not sufficient methods to deal with control flow, so we only choose arithmetic instructions for this paper. Although the instructions are not as diverse as high-level language in functionality, our vocabulary has included registers, memories, and constant numbers to make the story complete. Taken all these components together, it became a huge searching space which make it hard to scale up.
>
> As for previous work, we are glad to get to know more literature through your reviews. During our research, we try to cover as much literature as we can before deciding on the setting of assembly language. It’s no doubt that DSL is a good point not only in that many SOTA methods came from this research field, but also it already brought up practical ad-hoc solutions to real life scenario. Working on different settings, we hope by introducing assembly language into the program synthesis field, a joint effort from fields like programming language and deep learning can pour in consistent with a more general computer architecture.
>
> For the question about held out examples, we think this is really a good point, which is also mentioned by another reviewer. We have updated our work by doing more experiments on testing on held-out examples. In  the revised version, we use the same number of I/O pairs as before, but test all experiments with an unseen hold-out I/O pair. Although results from all models drop, AutoAssemblet still outperforms baselines reported. However, as stated in the updated figure, adding more I/O pairs does not help neural network recovering the correct program, which is quite interesting.
>
> As for previous work combining execution information and search method, I think we are using some intuitive method which have been explored by other works, like executing information(intermediate state in our case) and search method(MCTS and some variant in our case), which we cannot declare are our distribution. However, to make reinforcement learning work in the assembly generation is not trivial, because agent trained on data collected by itself can lead to serious distribution shift in a complex and non-convex state space like assembly code.So we adopted task resampling and policy sampling to further mitigate the problem.  Our generation process can also interact directly with gdb to get register memory value for planning, except for the problem that it is too slow for large scale training at current time.

---

### Decision · Program_Chairs · 2019-12-19

**Decision:**

Reject

**Comment:**

The authors consider the problem of program induction from input-output pairs.
They propose an approach based on a combination of imitation learning from
an auto-curriculum for policy and value functions and alpha-go style tree search.
It is a applied to inducing assembly programs and compared to ablation
baselines.

This paper is below acceptance threshold, based on the reviews and my own
reading.
The main points of concern are a lack of novelty (the proposed approach is
similar to previously published approaches in program synthesis), missing
references to prior work and a lack of baselines for the experiments.